# Indigenous Community Perspectives of Food Security, Sustainable Food Systems and Strategies to Enhance Access to Local and Traditional Healthy Food for Partnering Williams Treaties First Nations (Ontario, Canada)

**DOI:** 10.3390/ijerph18094404

**Published:** 2021-04-21

**Authors:** Ashleigh Domingo, Kerry-Ann Charles, Michael Jacobs, Deborah Brooker, Rhona M. Hanning

**Affiliations:** 1School of Public Health and Health Systems, Faculty of Health, University of Waterloo, Waterloo, ON N2L 3G1, Canada; rhanning@uwaterloo.ca; 2Cambium Indigenous Professional Services, Curve Lake, ON K0L 1R0, Canada; ka.charles@indigenousaware.com (K.-A.C.); m.jacobs@indigenousaware.com (M.J.); 3Ontario Ministry of Food, Agriculture and Rural Affairs, Guelph, ON N1G 4Y2, Canada; deborah_brooker@hotmail.com

**Keywords:** Indigenous health, food security, food sovereignty, food systems, sustainability, colonialism, community-based, participatory

## Abstract

In partnership with communities of the Williams Treaties First Nations in southern Ontario (Canada), we describe an approach to work with communities, and highlight perspectives of food security and sustainability, including priorities and opportunities to revitalize local food systems as a pathway to food security and food sovereignty. The objectives of our project were: (1) to build a shared understanding of food security and sustainability; and (2) to document community priorities, challenges and opportunities to enhance local food access. Utilizing an Indigenous methodology, the conversational method, within the framework of community-based participatory research, formative work undertaken helped to conceptualize food security and sustainability from a community perspective and solidify interests within the four participating communities to inform community-led action planning. Knowledge generated from our project will inform development of initiatives, programs or projects that promote sustainable food systems. The community-based actions identified support a path towards holistic wellbeing and, ultimately, Indigenous peoples’ right to food security and food sovereignty.

## 1. Introduction

Indigenous communities are increasingly driving the reclamation of traditional food systems (the term, *traditional food systems,* has been described as all foods identified within a particular culture that derive from local natural resources and includes sociocultural meanings associated with acquisition and use of such foods) as a means to enhance food security, food sovereignty and to sustain traditional food practices [1,2,3,4]. Indigenous holistic wellness has been supported by deep relations with the environment, including the natural resources it provides such as food and water for generations [3,4]. In Canada, however, the inequities faced by Indigenous peoples, tied to the loss of land, forced displacement from traditional territories and the history of colonization, have threatened the resilience of Indigenous food systems and impacted the right to food security and food sovereignty [1,2,3,4,5,6]. (Food sovereignty is the “*right of peoples to healthy and culturally appropriate food produced through ecologically sound and sustainable methods. It entails peoples’ right to participate in decision making and define their own food, agriculture, livestock and fisheries systems*” ([6]). The traditions and cultural practices to acquire, grow or prepare food have consequently been affected [1,4,7]. Challenges to ensuring food security have been further amplified by global climate change and environmental contamination of food systems, which have impacted access to, and quality of, land, water, plants and animal resources [8,9,10,11]. In addition to such processes that have threatened Indigenous food systems, rising costs of traditional food acquisition activities and local food production, along with the introduction of westernized food, has driven greater use of store-bought less nutritious food [12,13,14]. This nutrition transition has been tied to obesity and diet-related chronic diseases: a nutritional burden impacting Indigenous peoples worldwide [15,16,17,18].

In Canada, food insecurity has been felt most strongly by Indigenous peoples, recognized as First Nations, Inuit and Métis [12,13,14,15,16,17,18,19,20,21]. Food insecurity, an outcome and determinant of health, can range from worry about having enough food to limited access to, or reduced intake of, sufficient, safe, personally accepted, healthy food and even to hunger [21,22,23,24]. First Nations households on-reserve (Crown land designated by Canadian government for primary use of First Nations with registered Indian status [3]) in Ontario, have experienced a rate of food insecurity (41.7%) far greater than reported provincial household levels for the general population (11.6%) [24,25]. The extreme injustices exposed by the disproportionate rates of food insecurity impacting First Nations households highlight the need for community identified approaches to enhance food security in culturally safe and relevant ways. Sustainable food systems are central to long-term food and nutrition security [26] and extend beyond usual considerations for supportive economic, social and environmental conditions [27] to encompass cultural integrity.

In partnership with communities of the Williams Treaties First Nations in southern Ontario, Canada, we describe an approach to work with communities, and highlight perspectives of food security and sustainability, including priorities and opportunities to revitalize local food systems as a pathway to food security. Given the diverse challenges to food security described, momentum for the current work stemmed from broad considerations for the ecological health of the Lake Simcoe watershed under the Lake Simcoe Protection Plan and encompassed environment, climate change, agriculture and human health (the Lake Simcoe Protection Plan is part of the Lake Simcoe Protection Act, 2008, an Ontario government strategy to protect and restore the health of the Lake Simcoe watershed). Importantly, protecting and honouring harvesting rights for food, social and ceremonial purposes have long been recognized as essential to wellness by the Williams Treaties First Nations communities. Hence, creating opportunities to enhance and sustain local and traditional food access and knowledge has been identified as necessary to improve food security. As such, this participatory formative work was initiated and driven by communities to address food insecurity and enhance sustainability of food systems.

The objectives of our project were to: (1) build a shared understanding of food security and sustainability; and (2) document community priorities, challenges and opportunities to enhance local food access. The Williams Treaties First Nations, Cambium Indigenous Professional Services, the Ontario Ministry of Food, Agriculture and Rural Affairs, and the University of Waterloo coordinated and documented the process of data collection. Through the participatory process, we hope the formative work will initiate community capacity building and set the stage for further community-led action. The overall goal of this participatory formative work, is to inform a pathway of community driven solutions to strengthen food security and sustainable food systems, thereby promoting Indigenous people’s right to food and holistic wellness.

## 2. Materials and Methods

### 2.1. Project Team and Guiding Principles: Indigenous Communities and Academic Partnership

A project advisory committee with liaisons within each of the four partnering communities was established to guide project activities. Respectful and reciprocal relationships, approached through a decolonizing and participatory process, represents the foundational core of how this formative project was undertaken in ways that are culturally safe and ethical. The project team consisted of both Indigenous and non-Indigenous partners who engaged directly with participating Williams Treaties First Nations communities to identify culturally-relevant capacity building approaches to increase access to affordable, nutritious foods.

The project teams’ skills and knowledge positioned this participatory project to advance community-identified priorities and enhance access to sustainable local food systems. Project co-investigators from Cambium Indigenous Professional Services provided First Nations led expertise in environmental consulting and engineering services as members of Georgina Island First Nation and Curve Lake First Nation. Pre-existing relationships established with each of the partnering communities provided the opportunity to engage directly with community-identified expert advisors to continue ongoing food security and sustainability dialogues. Academic partners from the University of Waterloo supported by co-facilitating conversations with community advisors, evaluating and synthesizing findings and preparing project reports for funder and communities. Community members, identified through Cambium Indigenous Professional Services contacts and snowball approaches, were critical to the integrity of the process and relevance and utility of the findings.

Guided by the First Nations principles of Ownership, Control, Access and Possession (OCAP) [28], the project partners ensured engagement with communities was respectful, appropriate, beneficial and relevant to the communities involved. Information was only collected when informed assent was provided by participants, whose personal identification information remained anonymous. The OCAP principles applied helped to ensure integrity and meaningful engagement with communities to inform and control the direction of priorities identified throughout. Funding support was provided by the Ontario Ministry of Agriculture, Food and Rural Affairs as part of a broader commitment to the Lake Simcoe Protection Plan. Canada’s Tri-Council Policy Statement: Ethical Conduct for Research Involving Humans ([29], article 6.11, p. 77), indicates the “*initial exploratory phase, which is intended to establish research partnerships or to inform the design of a research proposal, and may involve contact with individuals or communities*” does not require full REB review (U Waterloo, Office of Research Ethics).

### 2.2. Partnering Communities of the Williams Treaties First Nations

The Williams Treaties First Nations are located in the province of Ontario, Canada within the Georgian Bay, Lake Simcoe and Lake Ontario watersheds of Treaty 20. The First Nations of the Williams Treaties comprise of the Chippewas of Georgina Island First Nations, Beausoleil First Nation, and Rama First Nation; and the Mississaugas of Curve Lake First Nation, Alderville First Nations, Hiawatha First Nation and Scugog Island First Nation. These communities, which include First Nations with non-registered and registered Indian status (Registered Indian status refers to an individual registered under the Canadian Indian Act [30]), are a diverse and growing population that strongly embrace a holistic relationship with the land, water and its local food systems. Communities of Georgina Island First Nation, Beausoleil First Nation, Curve Lake First Nation and Rama First Nation expressed interest in participating in this project as community partners.

The Beausoleil First Nation is situated at the Southern tip of Georgian Bay on Christian Island, Ontario. It is a small remote community, home to approximately 800 year-round residents, and has a membership of approximately 2214 people. The Island’s main access is by ferry transportation except, during the winter months, when access to the island is made over ice roads or hovercraft. As such, community members need to travel by vehicle and ferry to access the nearest store for groceries. The Island is also referred to as Chimnissing, which means “Big Island” in the Ojibway language. The First Nation consists of three Islands known as Christian, Hope and Beckwith, as well as 25 acres on the mainland at Cedar Point. Members predominately reside on Christian Island, however, several families live year-round on Cedar Point.

Curve Lake First Nation is also an Ojibway community, located 25 km northwest of Peterborough, Ontario. It has a diverse population of 2500, which includes First Nation members and non-First Nation members alike residing on territorial lands. There are 1918 registered status Indian members (1161 off reserve and 764 on reserve). The territory has a communal land base of 900 hectares, which consists of a mainland peninsula, a large island (Fox Island) and several other smaller islands located throughout the Trent Severn Waterway system. Curve Lake First Nation is a proud community known for its leadership and promoting Anishinabe culture. The health centre in Curve Lake, offers a range of programs and services including a healthy eating programming and a food bank. While there is access to convenience stores and a coffee shop, many must travel to Peterborough to shop at the nearest grocery store.

The Chippewas of Georgina Island First Nation is located both on and off the east shore of Lake Simcoe in the Region of York and is approximately 100 km north of the Greater Toronto Area. Georgina Island First Nation Reserve No. 33 consists of three separate islands and two mainland access points. The three islands (Georgina Island, Snake Island and Fox Island) are approximately 3 km off the southern shore of Lake Simcoe. The main population of the reserve resides on the largest island, Georgina Island, with approximately 90 households. The island’s land mass is approximately 15 km, which is 4.5 km long and 3.2 km wide with an area of 1415 hectors. Snake Island is approximately 135 hectors; and Fox Island is approximately 20 hectors, both of which are comprised of seasonal cottage residence and non-First Nation member residence. Travel by vehicle and ferry is needed to access the nearest store for groceries.

The Chippewas of Rama First Nation has been known as ‘the gathering place’, where travelers journeyed to trade, to seek counsel or medicines, and attend great meetings. Rama is now home to Casino Rama, a tourism attraction that brings many visitors to the area for gaming, shows, conventions, shopping and fine dining. Rama First Nation is approximately 1.5 h north of Toronto, Ontario. The community has over 2500 acres of interspersed land, nestled in “Ontario’s Lake Country” on the eastern side of Lake Couchiching. The community has over 1800 members including over 700 living off-reserve. Rama has a food bank and some local food and restaurant access but most travel to nearby Orillia (14 km) for groceries.

### 2.3. Project Design and Process

Applying a decolonizing theoretical perspective [31,32,33,34,35], we employed a community-based participatory research (CBPR) [33] methodological approach to advance the following formative project objectives: (1) to conceptualize food security and food sustainability from a First Nations perspective; and (2) to identify community-based approaches to enhance, protect and sustain local food systems in support of food security and sustainable food systems (Figure 1).

A participatory orientation emphasizes strong collaboration, non-hierarchical relations, and sharing of perspectives throughout the research process [36,37,38]. Given the partner centred and collaborative approach underpinning CBPR, it has increasingly been accepted as an appropriate methodology to engage with Indigenous communities as its relational component aligns with an Indigenous paradigm [35,39,40,41]. Applying CBPR within Indigenous contexts can facilitate space for Indigenous knowledge systems by shifting a focus on community values, perspectives and priorities [33,34,38,39,41].

To support communities with enhancing local food security and sustainability, the principles of CBPR were applied to facilitate the identification of community-led action and enablers for project planning. Further, CBPR in conjunction with Indigenous approaches to inquiry, akin to *two-eyed seeing* [41,42], was used to decolonize the research process and draw on the strengths and wisdom of Indigenous knowledge and ways of knowing. Formative work helped conceptualize food security and sustainability from a local community perspective and solidify interest within the four participating communities to inform and support community-led action planning in moving forward with their identified priorities.

### 2.4. Indigenous Approach to Inquiry

Recognizing the importance of telling stories in Indigenous culture, we used the conversational method to facilitate the sharing of reflections and experiences [33,34]. The conversational method is a dialogic approach to generating and gathering knowledge through oral storytelling and is congruent with Indigenous methodologies [33,34]. This was applied to facilitate community-based conversational/storytelling interviews involving semi-structured open-ended questions guided by the literature. This approach was undertaken with the aim of decolonizing the proposed work by enabling the creation of a safe space for partnering Williams Treaties First Nations community voices to be heard (Figure 1) [33]. As such, conversations were co-facilitated by an experienced community member to enable a safe environment for dialogue.

Using the conversational method, a series of community engagement meetings were held between December 2017 and March 2018 with four First Nation communities. Community liaisons in each partnering community, employed by the First Nation on climate change adaptation in their community, helped to identity advisors to engage in community dialogue sessions. Advisors with extensive knowledge of and expertise in environmental sustainability, food systems, traditional food practices, health, education and community development were invited to be engaged in the project. Academic partners, co-facilitated conversations with community expert advisors. Engagement sessions provided the opportunity to build relationships with respective community members and solicit input from the communities on perspectives of food security and food sustainability. In conversations with communities, informants identified priorities, challenges and opportunities for community-driven food-related projects to enhance food security and support sustainable local food systems.

Our Land, Building Capacity in Ontario First Nation Communities (Our Land) was a First Nations led and informed conference that was held on January 2018 at Rama, Ontario. This annual conference, organized by the project partners from Cambium Indigenous Professional Services, convened over 30 First Nations community representatives across Ontario.

The conference featured keynote presentations, panel presentations, participatory workshops, as well as an interactive digital survey to engage with conference delegates. On day one of the conference, we planned and facilitated a participatory workshop to explore ways to increase access to healthy food through the development of local food initiatives. The workshop featured a short video on food sustainability and food security, preliminary digital information gathering, an ice-breaker activity (trivia game and card game), facilitated break-out group discussions, and a dot-voting method to identify shared priorities. Conference attendees visiting the project booth were also invited to complete an optional survey to provide input on priorities and challenges to food security in their community. On day two of the conference, a digital survey of all delegates took place, based in part, on the findings of day one information gathering. All notes, flip charts and votes were collected.

The workshop guiding questions and both the paper-based and digital surveys were informed by knowledge gained from a review of the literature and community engagement sessions with partnering communities as a way to fill any knowledge gaps. In addition, early findings from the workshop helped to refine questions included in the digital survey on day two of the conference. The purpose of the workshop, the paper-based survey, and digital survey was to solicit input from Our Land Conference delegates on understandings of food security and food sustainability, and perspectives on community-level initiatives and capacity needed to achieve food security.

### 2.5. Data Analyses

Information gathered from the community-based conversational/storytelling interviews was transcribed and thematically analyzed [43,44]. Data used in this project are owned by communities as per First Nations Ownership, Control, Access and Possession guidelines (OCAP^TM^). In addition, notes from the participatory workshop at the conference and surveys were also analyzed to inform cross-cutting themes across all four partnering communities. Thematic analysis was used to systematically identify and organize patterns of meaning and experiences as described by Braun, Clarke and colleagues [43] and Cresswell and Cresswell [44]. The approach identified insights and patterns of meaning as they related to perspectives of food security and strategies to enhance local access to food. The code book and emergent themes were reviewed by the project team for refinement and to ensure inclusiveness and appropriate representation of community voices.

## 3. Results

### 3.1. Project Participants

Community key informants (*n* = 15) who were engaged in conversational/storytelling interviews (2–6 participants per community) included leaders in food security, environmental and energy sustainability, local food producers, and educators. Attendees of the conference included key partners and stakeholders working in the areas of climate change, environmental sustainability, energy, agriculture, public health, food security, and government. Conference delegates shared their perspectives through a participatory workshop (*n* = 65), and survey (*n* = 22 delegates), and large group feedback (*n* = 85). In addition to partnering communities of the project, the experiences and voices shared by First Nations community representatives across Ontario who attended the conference contributed to knowledge generation of priorities and opportunities for food security. The perspectives of key informants from each community on food security, food sustainability and associated challenges had common themes, though preliminary ideas for initiatives to strengthen local food systems were sometimes context specific. Given the small numbers, perspectives are not distinguished by respondent position or community.

### 3.2. Food Security

Food security was broadly understood from a holistic wellness perspective by all partnering communities. It was viewed as both an outcome and determinant of wellbeing. Participants highlighted the importance of access to food, knowledge of how to grow your own food, and the skills and resources to prepare food. Also emphasized was the importance that food procurement from the land, whether through hunting, fishing or gardening, should honour traditional practices and be carried out in ways that protect the environment.

Perspectives of individual and community food security were shared. Food security was commonly described as a way of life: promoting and maintaining overall wellness from a holistic perspective. Perspectives shared largely centred around access and use of traditional foods and store-bought food (non-traditional food sources). The ability to grow and produce food all year round and the opportunity to share food with others within the community was widely appreciated by community members.

Community advisor 1: *“I’ll give you my whole idea on it, on what would be the dream for me, which would essentially be people growing their own food. Maybe having greenhouses to supply the things you’re not getting from the summer. Having root cellars for storage. Having community freezers for those who hunt, fish, trap, but then also using the hide using the furs and putting that towards either making clothing or whatever it ends up being. But also knowing your medicines around, being able to identify trees, what parts of the plant are edible what parts are medicine.”*

Community advisor 2: *“I mean to me food security is everyone having access to healthy foods.”*

In addition, the value of having the knowledge and skills to undertake traditional food activities was expressed. This was inclusive of community reflections on the importance of practicing ceremony associated with traditional food acquisition activities, as well as being able to partner with other communities to acquire local foods within a community established trading system. These were recognized as essential to supporting community’s ability to provide its own food (e.g., grow, hunt, harvest, self-produce) for multiple generations.

Community advisor 4: *“Food is medicine. If you’re sick and have a cold, you don’t take vitamin C capsules, you boil a light batch of cedar tea.”*

### 3.3. Sustainable Food Systems

While food security was the primary topic of community engagement dialogues, the connection between sustainable food systems and food security was recognized by each community. Communities often discussed the importance of engaging in environmentally responsible food production as a way to protect the environment, while also growing and sustaining food within the community. In addition, communities expressed the importance of not relying on external food sources and also having the ability to control what goes into your food as a way to control the health of the community.

Community advisor 7: *“The ability to produce all year round would be nice. Trying to get the younger generations interested in fishing again, if we could tap into that a little.”*

### 3.4. Challenges to Food Security

Community identified challenges to food security centred around the impacts of climate change and environmental degradation, income, food knowledge and skills, and limited availability of healthy food options (Table 1).

#### 3.4.1. Climate Change and Environmental Degradation

Community members shared observations about changes in the abundance, distribution, and health of species (e.g., fish are smaller and have sores). Drivers of these changes identified include climate-related differences observed over time, from more drought in the summer (necessitating frequent fire bans) to longer growing seasons. Also shared was the observation of a growing number of invasive species that have altered plants and local food, including the presence of wildlife in the area. Some shared that environmental contaminants have also been responsible for impacting water and soil quality (road spraying oil and salts), including chemical waste dumped in waterways from a factory (reports of agent orange). As a result of these changes to local sources of food, communities have expressed having to travel further distances to hunt and fish because there’s not the same accessibility as there used to be.

Community advisor 6: *“…under our environmental assessment that we just had completed, one of the major issues on there that kind of held up the whole process for us would be the stuff that they spray on the road the dust control. And so that gives people a lot of concerns because you know we are so close, like a lot of the houses are so close to the road, our water sources [is] there and you know what’s running off into the water.*


*So [this] makes you think about, you know, if you were to have a garden, if each family had a garden, what’s running off into the gardens?”*


Community advisor 8: *“People travel further to hunt and fish to trap because it’s not the same like it used to be. So if they’re hunting muskrat there’s no muskrat left here really. There might be a few beavers.”*

#### 3.4.2. Income

Concerns were shared related to the high cost of food within the community, including high costs associated with travelling to grocery stores outside of the community. In addition, the costs to engage in traditional food activities related to travel to hunting areas, and access to tools or equipment to hunt was a concern. Amplifying these challenges is the dependency on social assistance due to low employment opportunities and rates in the community.

Community advisor 2: *“Food insecurity is huge for us. What is the root problem? It’s income basically.”*

Community advisor 6: *“I think also because we live in the cottage country that really determines the price especially meat and produce. I think sometimes we end up as during summer months like we are paying more for stuff that you know if you lived in the city you wouldn’t be paying that price. I think costs is the biggest factor.”*

#### 3.4.3. Traditional Food Knowledge and Skills

The loss of traditional knowledge, language and skills to engage in traditional food acquisition practices was emphasized as a barrier to food security. In particular, reflections were made on how the loss of traditional knowledge is tied to colonization and the Indian Act which have contributed to a lack of sovereignty and control over access to land to hunt, fish and gather (The Constitution Act (1876), Canada has the responsibility for ‘Indians and Lands Reserved for Indians’ under the Indian Act [3]. The Indian Act, which formally recognizes First Nations ancestry, is recognized as legislative authority that remains a source of internal colonization [3,30]). Shared was that such circumstances have not only led to less traditional food available to be consumed, but has also changed the perception and taste for traditional food.

Community advisor 6: *“I think we need to start with the younger generations because I think with the older generation we’re slowly losing that link to the past especially the ones that remember what it’s like way back when times were simpler, when people helped themselves, help each other, they didn’t have to rely with a lot of programs.”*

Emphasized, was the implication of limited traditional knowledge, in particular the lack of opportunities for traditional knowledge of food and culture to be passed down to younger generations on how to engage in community food gathering and harvesting practices, including knowing the location of hunting grounds and how to use tools or equipment to hunt.

Community advisor 9: *“I think there’s a gap that needs to be bridged somehow because we are losing our traditions and cultures within our own community. Because of that we are having to bring other people in that are teaching the same skills, but they are not our community’s skills.”*

### 3.5. Priorities to Enhance Food Security and Strengthen Local Food Systems

Priorities and actions were identified within each community to enhance food security (Table 1). These ranged from specific interests in food production, types of food, sustainability practices and developing food knowledge and skills, to community-based projects to implement in the short-term to support community interests and priorities.

#### 3.5.1. Access to Food

Access to grocery stores outside of the community was emphasized. Within each community, the importance of having access to means of transportation (either by bus, car or boat) was commonly expressed, as this was recognized as essential to accessing food, and especially healthy food of good quality.

Having a community-based travel system in place with appropriate storage was identified as an opportunity to support access to grocery stores outside of the community. For example, having a community-based bus that could accommodate large grocery portions and preserve freshness of vegetables in the summer was discussed.

Community advisor 5: *“…my one Aunt she won’t even buy bananas in the winter because they are brown by the time they get on the boat across the bay, but I think too, a lot of people like having to transport and so many people are on the boat. So if the ice starts building up then we use our passenger ferry only, so if you have 70 people riding a ferry then all their food and stuff like you kinda have to watch yourself.”*

Community advisor 6: *“But again it’s just our access has always been an issue because you know you have to bring stuff over. We do have farmers market in the summer, sometimes they are well received and sometimes they are not. It depends who’s bringing stuff over; sometimes it’s fresh, sometimes it’s not.”*

Also highlighted as key to supporting access to food, was ensuring sufficient income and employment opportunities within the community. Creating jobs within the community to support economic growth and security was emphasized.

Community advisor 2: *“And we know about [how] income and accessing affordable fresh produce in general, changes their body. Chronic disease goes down, you feel good. You’re not losing limbs because you’re eating sugary bread all the time.”*

#### 3.5.2. Availability and Use of Fresh, Nutritious Food

The importance of making fresh produce available and affordable for communities to access at wholesale costs was emphasized. Having food markets in the community with a variety of vendors of locally produced food and other essential products was identified as an action that could support greater access to fresh food. Having stable and reliable access to enough healthy and affordable food year-round without worry was broadly understood by communities as having secure access to food.

Support for health promoting activities to increase food literacy and knowledge of healthy foods and recipes was also identified as a priority that can support community in healthy eating practices. Developing incentives to have community members engaged at local markets and community-based food initiatives was also discussed.

Community advisor 7: *“I think with our community, I’ve heard from [a] few people that the issue too is education and equipment right. So, I mean, just finding out from people like, Ellen, you know getting that knowledge in order [to know] ‘where do I start?’. Some areas in our community are so sandy that you probably couldn’t get a garden, ‘so what do you have to do, okay fine you need to build up, so you have to build a crib right.’ ‘Who do I talk to?’ ‘Who has the skills to build this crib right?’*

Community advisor 3: *“I think it’s trying to find ways to provide fresh foods and vegetables. Cause like I said, they have this program called, the good food box, but that’s only catered towards certain people right; it’s not available to everybody.”*

An existing initiative within one of the partnering communities, Nourish, provides affordable access to local produce and market foods, was identified as an opportunity to support local food production and enhance community food security.

Community advisor 2: *“Nourish is a collaboration from Peterborough and public health from the YMCA which we are housed out of. We are a small group. We try to bring people into the system to help with system change…It’s all about food. We have people entering into our programs so we do cooking, we do growing, we have community meals, and we do advocacy work. Basically, the bottom line is to empower people to make decisions for themselves in order to make system changes.”*

Accelerating the uptake of existing projects within the community to support scale-up efforts, including support for local business to enhance traditional food supply (e.g., Nourish Project JustFood Boxes, restoration of wild rice beds, fish habitat improvements) [45,46] is an approach that can be taken and led by community to improve access to and availability of local, fresh and nutritious food.

#### 3.5.3. Restoring Community Connections to the Land and Traditions

Restoring and maintaining access to traditional food sources and distribution systems, such as wild rice production within the community, was emphasized. Communities shared the importance of promoting traditional food acquisition activities such as fishing, hunting and cultivating (e.g., wild rice production). This was discussed as an opportunity to re-establish connections to traditions, culture and the environment (e.g., water ceremony teachings, traditional medicines). This included promoting greater engagement with youth in traditional food teachings and encouraging younger generations to be advocates for local food projects and traditional food activities. In addition, utilizing social media platforms to engage youth and generate interest in learning more about traditional food and ways to contribute to food security in their community through gardening, gathering of traditional medicines, fishing, hunting, and harvesting and production (e.g., wild rice, maple syrup, net making), was emphasized. An approach identified to support this was having hunters and trappers from the community share their knowledge and skills to other members of the community (i.e., community-specific, not just northern FN community coming down to teach), including teachings of food as medicine. In addition, having a communal resource for members of the community to access tools and equipment for hunting was also identified as an opportunity to enable greater engagement in traditional food acquisition practices.

Community advisor 8: *“I think if we are able somehow to find a way to get the kids excited about farming, gardening, and fishing that we might see a difference, you know? Them bringing that home to their own parents and then getting them interested as well, but I think we need to offer some, I don’t know some ways to get them involved.”*

#### 3.5.4. Support for Locally Grown Food in the Community

Creating a sustainable food system, that can support healthy eating in the community and opportunities to achieve food security through approaches that enable local food production (e.g., community gardening), was discussed. Communities highlighted the importance of having policies that enable better management, protection and preservation of the land from pesticides/herbicides, and industry related activities. This was identified as an important process to ensure food safety and minimize food system exposure to environmental contaminants.

Regulations in place, with community input to support ongoing environmental assessment of water and soil quality, was identified as a priority to support local food production and a way to build legislative trust within the community. As invasive species can pose a potential threat to traditional food systems, programs and services in place that can remove invasive species were identified as an opportunity to support long-term environmental health in the community. Holding workshops that can enhance community members’ skills and knowledge to engage in local food production and activities that promote energy sustainability was also emphasized.

Community advisor 7: *“…well I think there is movement already to tell you the truth. From what I see from the people who have access to programs and their wants and their needs. I think even the younger are more [willing] now to try things. We’ve done canning workshops here. The only thing that I think of is if people were to grow their own food and they have all this extra produce. They get overwhelmed by okay, what do I do with all this kale? So, there’s education in that and how to do that or maybe have a barter system set up or whatever trading and things.”*

## 4. Discussion

The formative work undertaken in this participatory project aimed to facilitate the building of shared understandings of food security and food sustainability, including opportunities to strengthen access to nutritious and sustainable food systems. The Indigenous approaches utilized for knowledge generation and sharing helped to ensure appropriate engagement that centred the priorities of First Nations communities at the core. In addition, the findings will support community-led action planning and will be used to inform next steps for implementation. Knowledge generated from our project is intended to help develop initiatives, programs or projects that promote sustainable food systems. Such community-based action supports a path towards holistic wellbeing and, ultimately, Indigenous peoples’ right to food security and food sovereignty.

### 4.1. Strengthening Indigenous Food Security, Food Sustainability and Food Sovereignty

Food security was broadly understood from a holistic wellness perspective by all partnering communities. It was viewed as both an outcome and determinant of wellbeing. Participants highlighted the importance of access to food, knowledge of how to grow your own food, and the skills and resources to prepare food. Also emphasized was the importance of food procurement from the land, whether through hunting, fishing or gardening, should honour traditional practices and be carried out in ways that protect the environment.

Conceptualizations of food security underscored the importance of considering unique dimensions as related to access, availability and use of market and traditional food systems in efforts aimed at enhancing food security in Indigenous communities [21]. For Indigenous communities, this means a consideration of factors such as income and transportation that impact access to and use of market and traditional food; climate change and environmental degradation impacts on availability, use and access to traditional food and locally produced food; and how traditional food knowledge and skills, and connections to the land influence preparation and use of traditional and local food. In addition to such unique considerations of food security, an underlying theme was food sovereignty [5,6,47,48,49,50,51].

Food sovereignty is closely aligned with how partnering communities described food security, food sustainability and opportunities to improve access to food [47,48,49,50,51]. Underpinning community interests and priorities are actions that can support reclamation of access to land, revitalization of local food systems, reconnection with culture, traditional food and ways of knowing, and having greater control over ways to obtain healthy food within local food environments of the community. Such priorities align closely with what has been described as Indigenous food sovereignty.

Dawn Morison, for example, describes Indigenous food sovereignty as “*present day strategies that enable and support the ability of Indigenous communities to sustain traditional hunting, fishing, gathering, farming and distribution practices, the way we have done for thousands of years prior to contact with the first European settlers*” ([51], p. 98). Promoting food security and nutrition in a sustainable, culturally relevant and just way, as described by community members, is a window to enhancing Indigenous food sovereignty. Activities identified by communities that can improve greater access to food, restore connections and cultural ties to the environment, and support local food production, highlight their need to protect and restore the right to food and food sovereignty. As such, efforts to support revitalization of local food systems as a way to restore, protect and sustain community food practices is a community identified priority and opportunity to strengthen food security and food sovereignty.

### 4.2. Moving from Priorities to Community-Led Action

While partnering communities recognize the need to mobilize interests towards creating more sustainable and accessible food systems, moving from priorities to actions requires a level of readiness and capacity to plan, develop and implement initiatives. In addition, it requires working within the specific context of each community’s traditional and market food systems. Building on the momentum to better understand their local context as it relates to challenges and opportunities to strengthen access and availability of food, communities can begin thinking about how to create an environment for transformation and meaningful change. Existing implementation and change frameworks have highlighted the importance of understanding local context to build shared knowledge, assumptions and practices to define the change pursuits of interest as an initial step before effective implementation can take place [51,52,53].

The community-identified priorities from this formative work therefore present opportunities for change, whereby communities mobilize their interest for sustainable food systems, food sovereignty and food security by identifying and building on existing strengths within the community to enhance readiness and capacity for action. This may include identification of key system actors and champions, along with existing projects, initiatives and programs that can be scaled-up within the community. In addition, successes and lessons identified from existing or previous projects can also be leveraged. For example, an existing initiative, Nourish, was identified by community members as an opportunity for scale-up; extending its reach to nearby First Nations communities and incorporating traditional food could support their access to nutritious, culturally-relevant food. Strengthening relational capacities with key system actors has been identified as a critical process to support successful change pursuits [52,53]. Within the context of Indigenous communities, this is especially critical to align change efforts with community identified priorities and interests [54]. Indigenous scholars have continuously emphasized the importance of appropriate and respectful approaches to engage with communities to build shared understandings of knowledge and practices as required to support the implementation of programs or policies intended to improve Indigenous health and wellbeing [31,35,54].

Ongoing engagement with key community members from this preliminary work will be essential to identifying key system actors and champions that can lead and sustain implementation efforts and ensure success is achieved in the community. Furthermore, such engagement, driven by the community, will be fundamental to ensuring that actions, including partner-support of implementation strategies, are taken in a culturally safe way that best serves the interests of communities.

### 4.3. Limitations

Community-based participatory work, though well respected and recognized as an appropriate approach to working with Indigenous communities, is not without challenges or shortcomings. Short-term funding cycles may not support additional efforts to plan and implement community-identified projects of interest. While time and resource constraints limited the immediate uptake of the current project, partners continue to identify opportunities to utilize and leverage the findings from this project to inform other areas of existing or future funded projects.

## 5. Conclusions

We use Indigenous methodologies and western approaches to partner with communities and engage with community advisors to solicit input on priorities and opportunities to enhance food security, strengthen sustainability of food systems, and in doing so support the promotion of food sovereignty. The shared understanding of community perspectives, priorities, challenges and opportunities to strengthen food security will benefit participating Williams Treaties First Nations communities directly. Moreover, the approach taken in this formative initiative will help to inform other work to integrate robust Indigenous practices and approaches to community engagement critical to building shared knowledge and understanding as an initial step and requirement of implementation of programs intended to improve Indigenous health. Hence, the contributions made to understanding ways to support the wellbeing of partnering Williams Treaties First Nations by taking a community driven and participatory approach to identifying priorities and opportunities to revive local food systems, will promote greater food security, sustainability and food sovereignty that may resonate with other Indigenous communities.

The current research can serve as an important foundation for planning Indigenous community-based projects and initiatives to strengthen food security, create more sustainable food systems and work towards food sovereignty.

## Figures and Tables

**Figure 1 ijerph-18-04404-f001:**
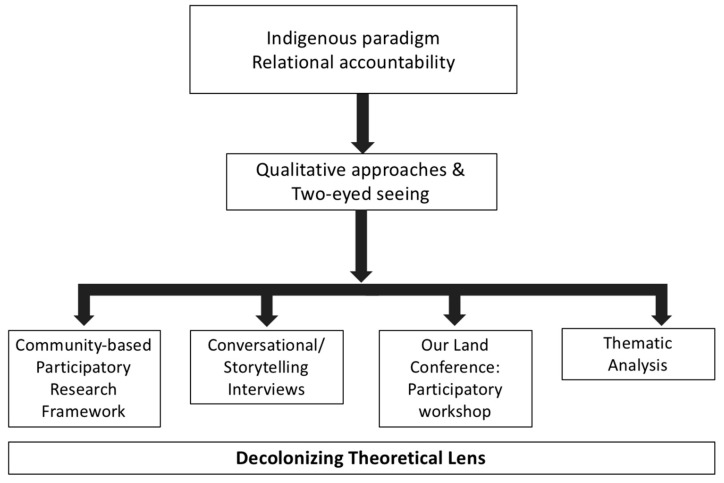
Overview of project design and Indigenous approach to inquiry.

**Table 1 ijerph-18-04404-t001:** Community perspectives of challenges and priorities to enhance food security and strengthen local food systems.

Challenges	Priorities
Climate change and environmental degradation	Access to food
Income	Availability and use of fresh, nutritious food
Traditional food knowledge and skills	Restoring community connections to the land and traditions
	Support for locally grown food in the community

## Data Availability

Data used are owned by communities as per First Nations Ownership, Control, Access and Possession guidelines (OCAP^TM^). Availability is dependent on reasonable request to the corresponding author and permission from the partnering communities.

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
