# Peer review of "Indigenous Community Perspectives of Food Security, Sustainable Food Systems and Strategies to Enhance Access to Local and Traditional Healthy Food for Partnering Williams Treaties First Nations (Ontario, Canada)"

_ijerph, 2021, doi:10.3390/ijerph18094404_

Round 1

Reviewer 1 Report

The Introductions sets the scene for what should be an exciting and topical paper. However, the manuscript is strongly let down by lack of details in the methodology and results sections. These need to be improved significantly to meet the requirements for rigour and repeatability in a study, as well as objectivity in assessment. In some instances, this calls for some restructuring of the paper

Author Response

Thank you kindly for your review and thoughtful suggested changes. Please see the attachment for a point-by-point response. 

Reviewer 2 Report

Presented to revive paper is a study on Indigenous community perspectives of food security, sustainable Food Systems and strategies to enhance access to local and traditional healthy food with Partnering Williams Treaties First Nations, Ontario, Canada. Generally. I find the paper well written and detailed, and the authors have sufficiently motivated the importance of the study. The authors note that food security issues may include concerns about having enough food to have limited access or reduced consumption of sufficient, safe, personally acceptable, healthy food, and even hunger.

Comments to consider:

Title (L2-5)

The paper has a suitable, detailed title.

Abstract (L13-25)

Well written and detailed. I did not note any major improvements to make.

Keywords (L26-27)

Consider including words which reflect terms relevant to the content, but not present in the title of the paper.

Introduction (L29-79)

The introduction is well written, but not very concise, but ultimately defines the purpose of the work.

Materials and methods (L81-236)

The conversational method was used as the basic method for generating and gathering knowledge in the roster of conversations with social counsellors.

Conclusions (L534-556)

It is well written and accurately reflects the main goals of the paper.

Author Response

Thank you for your feedback. Please see the attachment. 

Reviewer 3 Report

The issue is relevant and the paper is interesting and I can see why it can be of interest to different stakeholders. The paper is straightforward, well-written and well structured. Methods are appropriate and are adequately described.

Round 2

Reviewer 1 Report

Thank you for addressing my initial comments. I am satisfied with the responses. I have the following suggestions:

  • Ethical clearance - please include a footnote in the methods section to provide the explanation as other persons may also raise the same query even after the paper is accepted and published
  • Results section - I am still concerned that there is no visualisation of results in the entire section. For any thematic analyses, this is important and could be achieved using NVivo or free software such as VosViewer. Even a table to provide a synthesis of the results would have been helpful, either in the Results or Discussion section
  • Appendices - the idea these days is to make original datasets available, not so much for the reviewer, but for the greater good of science and also to test the reproducibility and repeatability of research. My suggestion is that add a statement as follows at the end of the manuscript, together with the Funding statements etc:

Data availability: data used in the study is available upon reasonable request from the corresponding author.

Author Response

Thank you for your feedback. Attached is our responses. Much appreciated for your thoughtful comments. 
